# Defining remission of type 2 diabetes in research studies: A systematic scoping review

**Mireille Captieux** *, **Regina Prigge**, **Sarah Wild**, **Bruce Guthrie**

Usher Institute of Population Health Sciences and Informatics, The University of Edinburgh, Edinburgh, United Kingdom

☯ These authors contributed equally to this work.
* mireille.captieux@ed.ac.uk

## Abstract

### Background

Remission has been identified as a top priority by people with type 2 diabetes. Remission is commonly used as an outcome in research studies; however, a widely accepted definition of remission of type 2 diabetes is lacking. A report on defining remission was published (but not formally endorsed) in Diabetes Care, an American Diabetes Association (ADA) journal. This Diabetes Care report remains widely used. It was the first to suggest 3 components necessary to define the presence of remission: (1) absence of glucose-lowering therapy (GLT); (2) normoglycaemia; and (3) for duration ≥1 year. Our aim is to systematically review how remission of type 2 diabetes has been defined by observational and interventional studies since publication of the 2009 report.

### Methods and findings

Four databases (MEDLINE, EMBASE, Cochrane Library, and CINAHL) were searched for studies published from 1 September 2009 to 18 July 2020 involving at least 100 participants with type 2 diabetes in their remission analysis, which examined an outcome of type 2 diabetes remission in adults ≥18 years and which had been published in English since 2009. Remission definitions were extracted and categorised by glucose-lowering therapy, glycaemic thresholds, and duration. A total of 8,966 titles/abstracts were screened, and 178 studies (165 observational and 13 interventional) from 33 countries were included. These contributed 266 definitions, of which 96 were unique. The 2009 report was referenced in 121 (45%) definitions. In total, 247 (93%) definitions required the absence of GLT, and 232 (87%) definitions specified numeric glycaemic thresholds. The most frequently used threshold was HbA1c<42 mmol/mol (6.0%) in 47 (20%) definitions. Time was frequently omitted. In this study, a total of 104 (39%) definitions defined time as a duration. The main limitations of this systematic review lie in the restriction to published studies written in English with sample sizes of over 100. Grey literature was not included in the search.

### Conclusions

We found that there is substantial heterogeneity in the definition of type 2 diabetes remission in research studies published since 2009, at least partly reflecting ambiguity in the 2009

**Data Availability Statement:** All relevant data cited within the manuscript is contained within the manuscript and supporting information file. Please access the raw data file at: doi: http://doi.org/10.17605/OSF.IO/F4H3R url: https://osf.io/f4h3r/.

**Funding:** MC is supported by a personal doctoral fellowship from the Chief Scientist Office Grant number: CAF 18/12 Website: https://www.cso.scot.nhs.uk The funder had no role in study design, data collection and analysis, decision to publish or preparation of the manuscript.

**Competing interests:** The authors have declared that no competing interests exist.

**Abbreviations:** ADA, American Diabetes Association; FPG, fasting plasma glucose; GLT, glucose-lowering therapy; PCOS, polycystic ovarian syndrome; PICOS, participants, interventions, comparisons, outcomes, and study design; PRISMA, Preferred Reporting Items for Systematic Reviews and Meta-Analyses; WHO, World Health Organization; 2-hr PG, 2-hour plasma glucose.

report. This complicates interpretation of previous research on remission of type 2 diabetes and the implications for people with type 2 diabetes. Any new consensus definition of remission should include unambiguous glycaemic thresholds and emphasise duration. Until an international consensus is reached, studies describing remission should clearly define all 3 components of remission.

## Systematic review registration

PROSPERO CRD42019144619

## Author summary

### Why was this study done?

- Remission of type 2 diabetes is very important to people with type 2 diabetes, but there is no single agreed definition of remission.

- Varying definitions of remission make research findings inconsistent and make remission difficult to effectively use as a target in clinical care.

### What did the researchers do and find?

- We systematically reviewed 178 research studies with an outcome of type 2 diabetes remission that had been published between 2009 and 2020 to evaluate how remission of type 2 diabetes had been defined.

- There were 96 unique definitions of type 2 diabetes remission, with substantial heterogeneity.

- A total of 93% of type 2 diabetes remission definitions stated that all glucose-lowering therapy (GLT) should be stopped. Moreover, 87% included a numeric glycaemic threshold, but the exact threshold and combination of tests to measure normal glycaemic levels varied considerably. In total, 61% did not include duration of stopping therapy or normal glycaemic levels as part of the definition of type 2 diabetes remission.

### What do these findings mean?

- The heterogeneity of type 2 diabetes remission definitions in the recent research literature highlights a fundamental problem for remission research.

- Synthesising the findings of studies of remission will be limited by variation in definition. Future meta-analyses should ideally use individual participant data.

- Categorising people into remission and non-remission groups in a way that is consistent with the current type 2 diabetes diagnosis guidelines is challenging and should be carefully considered before integrating remission with clinical type 2 diabetes management pathways.

- Researchers should always clearly and unambiguously state their definition of remission. There is a clear need for an internationally accepted and routinely implemented definition of remission.

## Introduction

Type 2 diabetes is a global health priority [1]. By 2045, an estimated 629 million people will be affected by diabetes, of whom 90% to 95% will have type 2 diabetes [1]. Drivers for the twin epidemics of obesity and diabetes lie in complex interactions between obesogenic environments, a biological tendency for weight gain, and an ageing population [2,3]. The rapid spread of obesity and diabetes worldwide has considerable health implications for the individual and major financial consequences for health services [3]. In 2015, the global economic burden of diabetes was estimated to be US$1.3 trillion [4], and total costs are estimated to rise to US$2.2 to $2.5 trillion by 2030 [4]. Type 2 diabetes has conventionally been considered a lifelong, progressive disease [5]. The concept of reversing type 2 diabetes by metabolic surgery was introduced in the early 1990s [6]. Buchwald and colleagues strengthened this concept by defining and demonstrating the resolution of clinical and laboratory manifestations of type 2 diabetes in their 2004 and 2009 systematic reviews and meta-analyses [7,8]. While such research groups did not publish any guidance on defining resolution of diabetes, their definitions of resolution developed for use in review and meta-analysis were widely adopted [9]. In 2009, a multidisciplinary expert group published a consensus report (2009 report) in Diabetes Care, an American Diabetes Association (ADA) journal to provide guidance on defining this concept [10], although this 2009 report does not represent the official ADA position [11]. The 2009 report recommends that the term "remission" should be applied to a chronic disease such as diabetes, rather than terms like "resolution." It broadly defines remission as "achieving glycaemia below the diabetic range in the absence of active pharmacologic or surgical therapy" [10] (p. 2134). Three specific types of remission are then explicitly defined: partial remission, complete remission, and prolonged remission (cure) (Table 1). Each remission type is composed of 3 components: (1) the absence of glucose-lowering therapy (GLT), (2) the achievement of a glycaemic threshold, and (3) a duration during which the other 2 components have to be sustained for remission to have occurred. The 2009 report is not to be confused with the annually updated ADA Standards of Care, which contains the official ADA position on the diagnosis of diabetes and prediabetes [12]; as yet, there is no official ADA position or internationally agreed consensus on the definition of remission [12].

Remission of type 2 diabetes is of increasing interest to professionals and patients [12]. Two recent trials in the United States of America and United Kingdom have demonstrated cost-effective remission of type 2 diabetes through intensive lifestyle measures such as very low-calorie diets [13,14]. These have reignited interest (particularly among people living with diabetes and their carers) [15] into whether remission might be a realistic goal for some people with type 2 diabetes and an additional strategy for health services. This issue is difficult to address when there is no official international consensus on how to define remission [16]. In the UK, the Primary Care Diabetes Society and the Association of British Clinical Diabetologists responded by publishing a consensus in 2019 which states that "remission can be achieved when a person with type 2 diabetes achieves 1. Weight loss; 2. HbA1c<48mmol/mol (6.5%) or FPG <7.0mmol/l (126mg/dL) on two occasions separated by six months; 3. Following

**Table 1. 2009 report recommendation for defining remission of diabetes and possible interpretations of recommendations (adapted from Buse, 2009) [10].**

| Remission term | GLT | Glycaemic threshold | Time (years) |
|---|---|---|---|
| Unspecified remission | Broadly defined in main text as "achieving glycemia below the diabetic range in the absence of active pharmacologic (anti-hyperglycemic medications, immunosuppressive medications) or surgical (ongoing procedures such as repeated replacements of endoluminal devices) therapy. A remission can be characterized as partial or complete." [10] (p. 2134) | | |
| Partial remission[a] | No active pharmacologic therapy (or ongoing procedures) | Hyperglycaemia below diagnostic thresholds for diabetes[d] | At least 1 year duration |
| Complete remission[b] | No active pharmacologic therapy (or ongoing procedures) | Normal glycaemic measures[d] | At least 1 year duration |
| Prolonged remission[c] | No active pharmacologic therapy (or ongoing procedures) | Normal glycaemic measures[d] | At least 5 years duration |

a Partial remission is specifically defined as "Sub-diabetic hyperglycaemia (HbA1c not diagnostic of diabetes <48mmol/mol (<6.5%), fasting glucose 5.6–6.9 mmol/l (100–125 mg/dL)) of at least 1 year's duration in the absence of active pharmacologic therapy or ongoing procedures." [10] (p. 2134)

b Complete remission is specifically defined as "A return to 'normal' measures of glucose metabolism (A1C in the normal range, fasting glucose <5.6 mmol/l (<100 mg/dl)) of at least 1 year's duration in the absence of active pharmacologic therapy or ongoing procedures." [10] (p. 2134)

c Prolonged remission is specifically defined as "Complete remission that lasts for more than 5 years and might operationally be considered a cure." [10] (p. 2134)

d There is ambiguity in terms of whether remission requires HbA1c<threshold AND FPG<threshold or whether the logic is HbA1c<threshold OR FPG<threshold (see footnotes a–c where the join between HbA1c and FPG is a comma rather than a logical operator).

FPG, fasting plasma glucose; GLT, glucose-lowering therapy.

complete cessation of all GLT." [17] (p. 74). The International Diabetes Federation discusses remission in the context of bariatric surgery in their 2017 clinical practice recommendations for primary care, where they state "Remission is defined by most guidelines as an HbA1c below 6% (42mmol/mol) without medication for 6 months or more" [18] (p. 21). Consensus discussions are ongoing as part of the response to calls for an international consensus on definition [16]. However, it is unclear how remission is currently being interpreted in research literature. The aim of this study is to systematically review how remission of type 2 diabetes has been defined by observational and interventional studies since the publication of the 2009 report, in order to contribute to developing future consensus on defining type 2 diabetes remission.

## Methods

### Search strategy and selection criteria

We adapted the Preferred Reporting Items for Systematic Reviews and Meta-Analyses (PRISMA) statement for scoping reviews to report our systematic review of observational and interventional studies [19] (S1 Table). We included observational and interventional studies involving at least 100 participants with type 2 diabetes in their remission analysis, which examined an outcome of type 2 diabetes remission in adults ≥18 years old, and which had been published in English since 2009 (when the 2009 report was published) [10]. These restrictions were decided a priori and were expected to provide a sufficient number of papers specifically focussing on the effect of the 2009 report in reasonably large research studies within our time and resource constraints (S2 Table). Web content or materials produced outside the traditional academic publishing were not searched as we were interested in how remission was defined in manuscripts that had been peer-reviewed in academic publication and distribution channels. We excluded case reports, systematic reviews, protocols, or reviews. Studies focussing on prediabetes, impaired glucose tolerance, impaired fasting glucose, gestational diabetes, maturity onset diabetes of the young, steroid-induced diabetes, or type 1 diabetes were excluded as beyond the scope of this review (S2 Table).

Patients or the public were not involved in the design, conduct, reporting, or dissemination plans of our research. The Diabetes UK–James Lind Alliance Priority Setting Partnership has recently identified remission as the top shared priority among people living with diabetes and their carers, healthcare professionals, and black and minority ethnic groups [15].

We searched MEDLINE, EMBASE, CINAHL, and the Cochrane Library from 1 January 2009 to 18 July 2020 (S2 Table). The participants, interventions, comparisons, outcomes, and study design (PICOS) search strategy [20] was developed with a senior librarian and combined terms for "remission," AND "type 2 diabetes," AND "weight loss strategies," AND "limits specified" (human patients, English language) (S3 Table). Two reviewers (MC and one of BG, SW, and RP) independently screened all title and abstracts and all full texts against our inclusion criteria using Covidence systematic review software (Veritas Health Innovation, Melbourne, Australia). MC and RP independently extracted data in duplicate to a piloted Microsoft Excel extraction form. In all cases, disagreement was discussed, and a third reviewer was involved to resolve these if needed. The systematic review protocol was prespecified and prospectively registered (PROSPERO registration: CRD42019144619).

## Outcomes

The primary outcome was the definition of remission used in each study irrespective of the term(s) used to name the outcome (partial, complete, or prolonged remission and synonyms such as resolution, cure, or reversal). Since an individual study could measure remission in more than 1 way, we extracted data for all distinct definitions used in each paper.

## Data analysis

For each definition of remission given by an underlying study, we sought to specify how each of the 3 components of remission defined by the 2009 report was operationalised (absence of GLT, normoglycaemia, and duration of 1 or both of absence of GLT and normoglycaemia), irrespective of whether they referenced the 2009 report [10]. For each definition, we recorded whether any components were omitted and quantified the heterogeneity within each component by counting the number of unique ways each component was defined. We mapped every definition extracted to the 3 types of remission initially introduced by the 2009 report (partial remission, complete remission, and prolonged remission) based on the term used by the authors. Where it was not possible to map to 1 of the 3 types, for example, authors used general terms such as remission or resolution, we categorised a definition as unspecified remission. We also counted the number of unique definitions across all definitions, regardless of the term used by authors. We examined how the 2009 report defined remission and listed feasible interpretations for each definition. We calculated the proportions of studies that cited the 2009 report, attempted to quantify fidelity to the 2009 report, and whether studies justified their particular interpretation. Assessing the methodological quality and risk of bias of included studies in terms of the effect of their intervention on remission was not relevant to our aims or outcomes. Assessing whether poor reporting of remission definitions was linked to high risk of bias was outside the scope of this preliminary exploration. Therefore, we did not undertake risk of bias assessment of the underlying studies in relation to the analysis they did.

## Results

The search identified 8,966 citations, and 6,772 title and abstracts underwent screening after removal of duplicates. After screening of 381 full texts against inclusion criteria, 178 studies from 33 countries met the inclusion criteria (Fig 1), comprising 164 cohort studies, 11 randomised controlled trials, 2 cluster randomised trials, and 1 cost-effectiveness model (S4 Table).

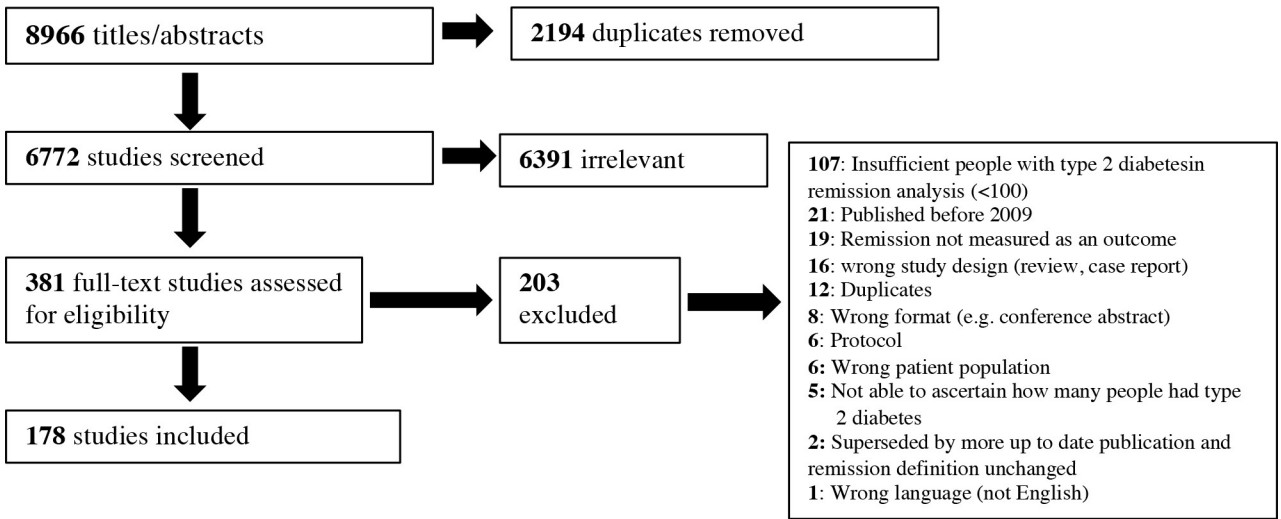

**Fig 1. PRISMA diagram showing selection of studies.** PRISMA, Preferred Reporting Items for Systematic Reviews and Meta-Analyses.

Surgical interventions were the focus of 164 (93%) studies compared to 8 (4%) pharmacological and 5 (2%) lifestyle interventions. One epidemiological study investigated the incidence of remission in adults in a diabetes registry [21]. The 2009 report was referenced in 70 (39%) studies and in 121 (45%) definitions. A total of 34 (19%) studies used an alternative or additional reference. Of the 18 alternative references, different versions of the ADA diagnostic guidelines for diabetes [12] or Brethauer and colleagues [22] were used most frequently (S4 Table). In total, 81 (46%) studies did not directly reference an existing guideline or paper.

## Variation in nomenclature and definition of remission

A total of 177 (99%) studies included 1 or more explicit definitions of remission, using 10 different terms: "partial remission," "complete remission," "prolonged remission," "sustained remission," "sustained remission of hyperglycaemia," "persistent remission of hyperglycaemia," "any remission of hyperglycaemia," "remission," "resolution" and "cure" (S4 Table). We combined definitions named as remission, remission of hyperglycaemia, and resolution into a single category (unspecified remission) and definitions named as prolonged remission and cure into a single category (prolonged remission) to create 4 groups. There were 266 distinct definitions extracted, since 61 (34%) studies examined more than 1 remission type (for example, separately defining partial remission and complete remission). Almost half of definitions were categorised as unspecified remission (124 definitions (47% of all definitions)) compared to the more specific terms partial remission (57 definitions (21%)), complete remission (64 definitions (24%)), and prolonged remission (21 definitions (8%)) (Table 2).

## Glucose-lowering therapy

The GLT component of remission definitions was the most consistently defined. In total, 254 (95%) definitions included a GLT component, with 247 (93%) requiring the absence of GLT and 8 definitions (3%) allowing continued use of GLT (either metformin or "some GLT") in their definition of remission (Table 2, Fig 2) [23–29]. One study specified that metformin prescribed for a non-type 2 diabetes indication, e.g., polycystic ovarian syndrome (PCOS), need not be stopped for remission to be achieved [29].

**Table 2. Definition of remission in terms of GLT, time, and glycaemic threshold components for all 266 definitions of remission cited[a].**

| | All 3 components defined[b] n (%) | GLT | Glycaemic threshold | Time | |
|---|---|---|---|---|---|
| | | Absence of GLT specified n (%) | Explicit numeric threshold specified n (%) | Defined cross-sectionally[c] n (%) | Defined longitudinally[d] n (%) |
| Unspecified remission 124 definitions (57 unique) | 45 (36.3) | 116 (93.5) | 95 (76.6) | 19 (15.3) | 34 (27.4) |
| Partial remission 57 definitions (22 unique) | 34 (59.6) | 49 (86.0) | 54 (94.7) | 9 (15.7) | 28 (49.1) |
| Complete remission 64 definitions (25 unique) | 38 (59.4) | 61 (95.3) | 62 (96.9) | 12 (18.8) | 27 (42.2) |
| Prolonged remission 21 definitions (11 unique) | 21 (100) | 21 (100) | 21 (100) | 6 (28.6) | 15 (71.4) |
| Total: 266 definitions (96 unique) | 138 (51.9) | 247 (92.9) | 232 (87.2) | 46 (17.3) | 104 (39.1) |

a Studies can contribute more than 1 definition of remission, for example, because they measure both complete remission and partial remission.

b Glycaemic threshold had to be defined with an explicit numeric threshold.

c Assessing remission at one moment in time such as at 1 year follow-up after an intervention.

d Assessing remission after criteria have been met for a duration of time.

GLT, glucose-lowering therapy.

## Glycaemic thresholds

In this study, a total of 232 (87%) definitions specified numeric glycaemic thresholds (Table 2), but with considerable heterogeneity. There were 25 unique numeric definitions of glycaemic threshold in unspecified remission definitions, 8 in partial remission, 11 in complete remission, and 9 in prolonged remission (Fig 2, Table 3). HbA1c was used more often than fasting plasma glucose (FPG) and 2-hour plasma glucose (2-hr PG). Of the 232 definitions that explicitly specified a numeric glycaemic threshold, 105 (45%) were based on HbA1c alone, and HbA1c<42 mmol/mol (6.0%) was used most frequently in 47 (20%) definitions. HbA1c<48 mmol/mol (6.5%) with or without normoglycaemic interpretations of FPG (103 definitions) and HbA1c<42 mmol/mol (6.0%) with or without normoglycaemic interpretations of FPG (89 definitions) were the most frequently used thresholds (S5 Table). It was not clear whether both or either of FPG and HbA1c needed to be attained in 23 definitions (S5 Table). There were 27 unique ways to categorise normoglycaemia (when all categories of remission type were merged, regardless of the term used by the authors) due to variation in the glycaemic thresholds for each of HbA1c, FPG, and 2-hr PG and also the combination of glycaemic tests specified (S5 Table).

## Time

The time component was the most poorly reported component of remission definitions. It was always specified in definitions of prolonged remission but not specified in 72 (58%) of unspecified remission definitions, 20 (35%) of partial remission definitions, and 25 (41%) of complete remission definitions (Table 3, Fig 2). When specified, the time periods defining duration of normoglycaemia ranged from 30 days to 5 years (Fig 2), with 17% of definitions defining time cross-sectionally (e.g., assessing remission at one moment in time such as at 1 year follow-up after an intervention) rather than longitudinally (e.g., absence of GLT and glycaemic threshold definitions met for a duration of 1 year) (Table 2). Eleven studies (13 definitions) specified a time period or follow-up of less than 1 year in their definitions [13,23,25,30–37] (Fig 2). We found 13 unique definitions for the time component (when all categories of remission type

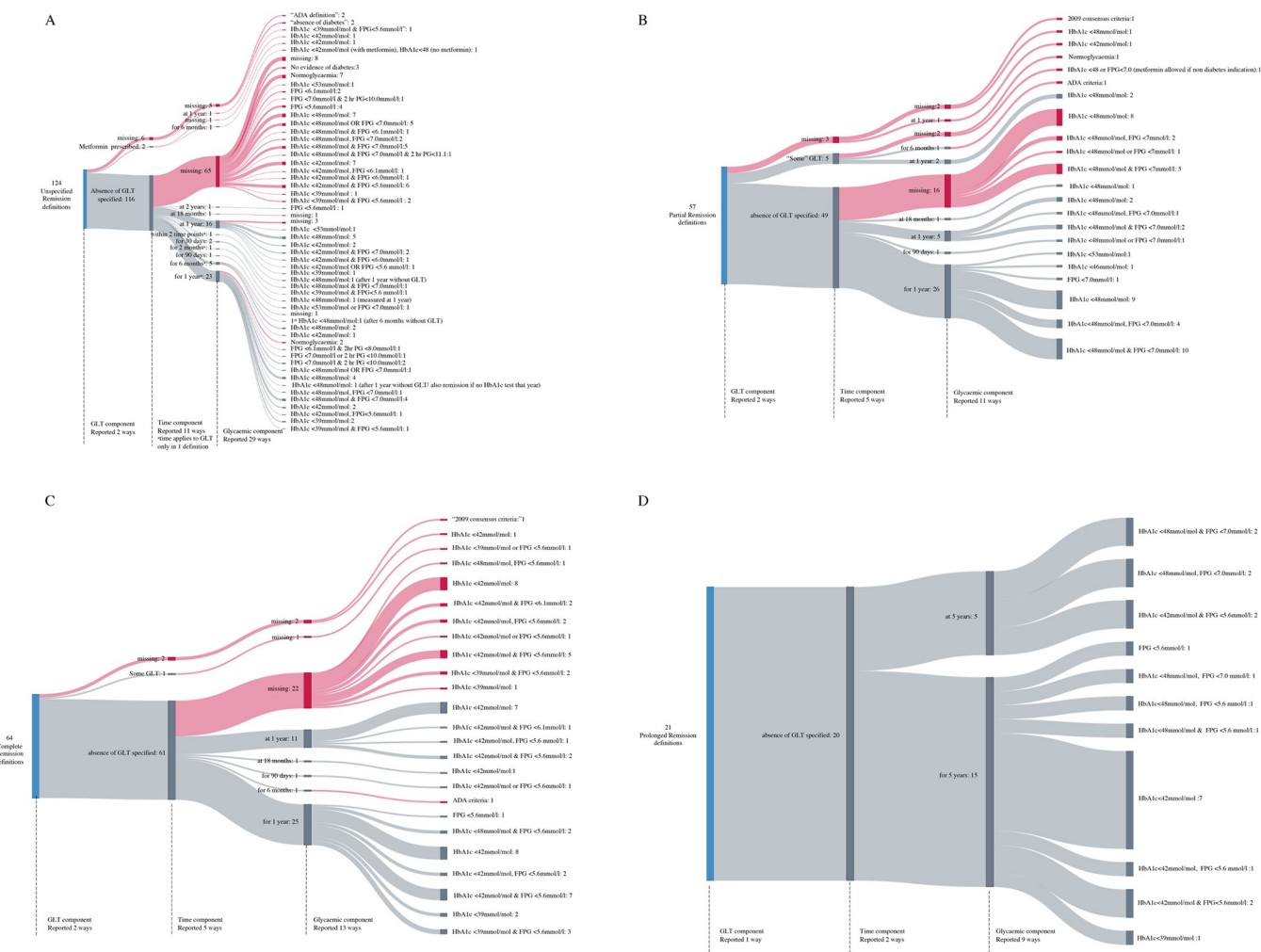

**Fig 2. Sankey diagrams showing heterogeneity in definitions of remission.** (A) Unspecified remission. (B) Partial remission. (C) Complete remission. (D) Prolonged remission. Red indicates a definition that has 1 or more missing (i.e., undefined or ambiguously defined) component. GLT, glucose-lowering therapy.

were merged, regardless of the term used by the authors) (Fig 2, Table 3). This was due to variation in the period of time specified, definition of time cross-sectionally or longitudinally, and applying a different definition of time to the other 2 remission components (Fig 2).

## Unique definitions

Overall, there were 96 unique definitions of remission. There were 57 unique definitions of unspecified remission, 22 unique definitions of partial remission, 25 unique definitions of complete remission, and 11 unique definitions of prolonged remission (Table 2, Fig 2). For each remission category, heterogeneity in defining glycaemic thresholds was the most frequent driver of total heterogeneity (Table 3, S1–S4 Figs).

## Discussion

Since the publication of the 2009 report, our analysis of 178 studies conducted in 33 countries identified substantial heterogeneity in how remission is named and defined. We identified 96

**Table 3. Unique definitions for remission in terms of GLT, glycaemic threshold component, and time components for all 266 definitions of remissions cited.**

| | No. (%) of definitions which did not clearly specify each component | GLT component No. of unique definitions | Glycaemic threshold No. of unique numeric definitions | Time component No. of unique definitions | Full definition No. of unique definitions |
|---|---|---|---|---|---|
| Unspecified remission (*n* = 124 definitions) | GLT absence 6 (4.9) | 2 | 25 | 11 | 57 |
| | Glycaemia 29 (23.4)[a] | | | | |
| | Time 72 (58.1) | | | | |
| Partial remission (*n* = 57 definitions) | GLT absence 3 (5.3) | 2 | 8 | 5 | 22 |
| | Glycaemia 3 (5.3)[b] | | | | |
| | Time 20 (35.1) | | | | |
| Complete remission (*n* = 64 definitions) | GLT absence 2 (3.1) | 2 | 11 | 5 | 25 |
| | Glycaemia 2 (3.1)[c] | | | | |
| | Time 25 (40.6) | | | | |
| Prolonged remission (*n* = 21 definitions) | GLT absence 0 | 1 | 9 | 2 | 11 |
| | Glycaemia 0 | | | | |
| | Time 0 | | | | |
| Total unique definitions (regardless of remission term)[d] | GLT absence 11 (4.1) | 3 | 27 | 13 | 96 |
| | Glycaemia 34 (12.8) | | | | |
| | Time 117 (44.0) | | | | |

a A total of 13 (10.5%) definitions were missing, and 16 (12.9%) definitions were ambiguous as they did not state a numeric threshold.

b A total of 3 (5.3%) definitions were ambiguous as they did not state a numeric threshold.

c A total of 2 (3.5%) definitions were ambiguous as they did not state a numeric threshold.

d Unique definitions merged across categories. Due to overlap in unique definitions when subcategories were merged, total is less than the sum of unique categories in each subcategory of remission.

GLT, glucose-lowering therapy.

unique definitions of remission that reflected heterogeneity in the 3 components relating to GLT (3 unique explicit definitions), glycaemic thresholds (27 unique numeric definitions), and time (13 unique explicit definitions), as well as heterogeneity in different combinations of these 3 components in each definition. The 2009 report was the most widely used guide in defining remission and was referenced in 121 (45%) definitions.

To our knowledge, this is the first review to systematically categorise and quantify heterogeneity of remission definitions in recent research. Previous studies have shown that the choice of glycaemic thresholds in remission definitions markedly affects the estimates of the proportion of people in remission [9,38–42]. Despite a variety of strategies to manage heterogeneous definitions of remission, systematic reviews have acknowledged the limitations of summarising data from primary remission studies and interpreting pooled estimates [43–46]. Our findings confirm and quantify the substantial heterogeneity and ambiguity in defining the glycaemic component of remission. We additionally identify that the inconsistent definition of time creates further heterogeneity.

Strengths of the study include systematic searching and that all screening and extraction was carried out independently by 2 reviewers. We used the PRISMA statement for scoping reviews to ensure clarity of reporting [19]. We had to adapt traditional PRISMA recommendations for a review assessing and evaluating a particular aspect of methodology (how remission was defined) rather than a traditional outcome-focussed systematic review. Potential limitations include the restriction to studies with sample sizes of over 100 and not searching for grey literature or performing a forward citation search. The review still included 178 studies with considerable heterogeneity in remission definition, but these limitations may have introduced bias. For example, smaller studies may have been carried out by practising clinicians who may

have used different definitions of remission to large research studies. However, the likely direction of any bias is towards underestimating true heterogeneity in defining remission, so we do not think our conclusions would change. In restricting to English-only manuscripts, we are potentially limited in our ability to draw conclusions on international practice; however, our included studies still demonstrate a good distribution of countries, so this decision may not have unduly biassed the outcome of the study (S4 Table). In the registered protocol, 1 objective was to evaluate fidelity to the 2009 report in studies citing it. However, while carrying out the review, we identified that the glycaemic component of the 2009 report is itself ambiguous in terms of (1) whether both or either FPG and HbA1c measures are required to determine normoglycaemia; (2) what constitutes normal glycaemic thresholds for complete remission and prolonged remission; and (3) whether remission is defined as a term in its own right. This precluded a meaningful analysis of fidelity (S6 Table). Some studies responded to this ambiguity by discussing their interpretation and implementation of the 2009 report remission definitions [21,41,47,48], while others [40,49–60] did not clarify whether both or either of FPG and HbA1c were needed. The ambiguity in the 2009 report is likely to at least partly underlie the heterogeneity we observed in the definition of remission.

The findings of this review have implications for research and identifies issues for future consensus groups to consider. The 2009 report authors stated that they hoped their recommendations would stimulate discussion, implying an expectation that their proposed remission definitions would evolve. A persisting problem is the difficulty in applying discrete binary terms such as "remission" to a chronic disease characterised by glycaemia, which is a continuous parameter [10,61]. Diagnostic thresholds for type 2 diabetes or prediabetes/intermediate hyperglycaemia have shifted in response to different interpretations of the association between glycaemia and vascular complications since the World Health Organization (WHO) first created a framework to diagnose diabetes in 1965 [62]. This review highlighted a tendency or preference in the research literature to use the word "remission" without qualification rather than "partial," "complete," or "prolonged remission." The complexity in defining remission and creating partial and complete subcategories mirrors the complexity in defining diabetes and prediabetes [12]/intermediate hyperglycaemia [62]. Complete remission mirrors the diagnosis of prediabetes which is in itself controversial and is not internationally consistent. We therefore suggest the alternative terms "remission of type 2 diabetes" (instead of "partial remission") and "remission of prediabetes" (instead of "complete remission") to make this clear. Creating and defining a remission of prediabetes category will be highly challenging (Table 4), and it may be worth focussing on a consensus for remission of type 2 diabetes in the first instance. There is an argument that combining multiple patient characteristics in a risk prediction model may be an alternative approach that sidesteps the issue of creating and defining a remission state and focusses instead on minimising future vascular complications rather than attaining remission [61]. However, until such tools are created, clinicians rely upon naming and classifying disease states that match diagnostic classification and coding schemas [63]. Future systematic reviews and meta-analyses would ideally use meta-analysis of individual participant data to apply a consistent definition of remission to make best use of existing data.

The first step in translating research to clinical practice is to provide clinicians with guidance on diagnosing type 2 diabetes remission. Given the fundamental difficulties in categorising type 2 diabetes, thought must be given to how remission guidelines maintain consistency with diabetes diagnostic guidelines. Type 2 diabetes is diagnosed by HbA1c, or FPG, or 2-hr PG. These continuous measures of glycaemia are categorised using the following thresholds: HbA1c $\geq$48 mmol/mol (6.5%), FPG $\geq$7.0 mmol/l (126 mg/dL), and 2-hr PG $\geq$11.1 mmol/l (200 mg/dL). As a consequence of diabetes diagnostic guidelines, the logical operator between HbA1c, FPG, and 2-hr PG would ideally be AND. This would avoid the situation of a person

**Table 4. Proposed alternative terms for remission of type 2 diabetes and remission of prediabetes/intermediate hyperglycaemia.**

| 2009 report category | Proposed category | Possible definition for discussion |
|---|---|---|
| Partial remission | Remission of type 2 diabetes | 1. Absence of GLT required<br>Consensus required on GLT prescribed for non-diabetes indications, e.g., metformin for PCOS<br>2. Normoglycaemia thresholds for diabetes diagnosis as defined by WHO and ADA applied in reverse:<br>• HbA1c<48 mmol/mol (6.5%) [12,64] AND<br>• FPG<7.0 mmol/l (126 mg/dL) [12,62] AND<br>• 2-hr PG<11.1 mmol/l (200 mg/dL) [12,62]<br>Consensus is required on whether all available glycaemic tests must be normal given ADA and WHO diagnostic guidelines or whether remission can be diagnosed based on the single most appropriate test for the patient (e.g., HbA1c, FPG, or 2-hr PG). If only 1 test is required for remission, how to manage conflicting glycaemic results, e.g., HbA1c<48 mmol/mol and FPG>7.0 mmol/l.<br>3. Absence of GLT and glycaemic thresholds both met for a "duration" of time<br>Consensus is required on the duration required, e.g., 6 months or 12 months. |
| Complete remission | Remission of pre-diabetes/ intermediate glycaemia* | 1. Absence of GLT required<br>Consensus required on GLT prescribed for non-diabetes indications, e.g., metformin for PCOS<br>2. Normoglycaemia thresholds used for prediabetes (ADA) or intermediate glycaemia (WHO) applied in reverse:<br>• HbA1c threshold is disputed<br>(a) ADA: HbA1c <39 mmol/mol (5.7%) [12]<br>(b) WHO: insufficient evidence [64]<br>• FPG threshold is disputed<br>(a) ADA: <5.6 mmol/l (100 mg/dL) [12]<br>(b) WHO: <6.1 mmol/mol (110 mg/dL) [62]<br>• 2-hr PG<7.8 mmol/l (140 mg/dL) [12,62]<br>Consensus required on thresholds and whether all available glycaemic tests must be normal or whether remission can be diagnosed based on the single most appropriate test for the patient<br>3. Absence of GLT and glycaemic thresholds both met for a duration of time<br>Consensus is required on the duration required, e.g., 6 months or 12 months. |

* Prediabetes itself is a disputed concept, and a consensus definition for a remission of prediabetes/intermediate hyperglycaemia is likely to be very challenging. WHO does not use the term prediabetes and suggests "intermediate glycaemia" which can include (1) impaired fasting glucose; FPG 6.1–6.9 mmol/l (110–124 mg/dL) and (if measured) 2-hr PG<7.8 mmol/l (140 mg/dL) and (2) impaired glucose tolerance; FPG<7.0 mmol/l (126 mg/dL) and 2-hr PG ≥7.8 (140 mg/dL) and <11.1 mmol/l (200 mg/dL)) [62] (p. 3)

ADA, American Diabetes Association; GLT, glucose-lowering therapy; PCOS, polycystic ovarian syndrome; WHO, World Health Organization; 2-hour plasma glucose, 2-hr PG.

diagnosed in remission from their type 2 diabetes based solely on an HbA1c<48 mmol/mol (6.5%) (one of the most commonly used definitions of remission) while simultaneously meeting criteria for diabetes diagnosis if their FPG was over 7.0 mmol/l (126 mg/dL) [12,64]. However, achieving remission in all 3 glycaemic measures has substantial time and cost implications. This definition would be impractical to implement and would not be clinically appropriate in all patients (e.g., HbA1c is an inaccurate measure of glycaemia in certain patients and not easily available in all countries [64]) (Table 4).Therefore, using either HbA1c or FPG or 2-hr PG approach may be the best compromise to define remission of type 2 diabetes. Potential inconsistencies between diabetes diagnosis and remission diagnosis should, however, be specifically addressed in guidelines to avoid causing confusion among clinicians and patients.

People with type 2 diabetes have prioritised remission of type 2 diabetes and want more guidance about how to achieve remission and its implications [15]. This systematic review focusses on how remission has been defined in the recent literature. In doing so, we have demonstrated diverse multiple definitions of remission. If the concept of type 2 diabetes remission is to be pursued and implemented in clinical practice, then proceeding without a widely adopted consensus remission definition will be confusing for patients and clinicians alike. Heterogeneity also impedes research reproducibility, building knowledge, and the provision of

clear guidance to people with type 2 diabetes. This review supports the need for an international consensus definition of remission to guide both research and clinical practice. In the meantime, any research study using "remission" as an outcome should unambiguously report and justify their definition in terms of absence of GLT, glycaemic thresholds, and time.

## Supporting information

**S1 Fig. Sankey diagrams to illustrate heterogeneity for each individual component of the remission definition: Glucose lowering therapy, time, and glycaemic component for unspecified remission.**
(PPTX)

**S2 Fig. Sankey diagrams to illustrate heterogeneity for each individual component of the remission definition: Glucose lowering therapy, time, and glycaemic component for partial remission.**
(PPTX)

**S3 Fig. Sankey diagrams to illustrate heterogeneity for each individual component of the remission definition: Glucose lowering therapy, time, and glycaemic component for complete remission.**
(PPTX)

**S4 Fig. Sankey diagrams to illustrate heterogeneity for each individual component of the remission definition: Glucose lowering therapy, time, and glycaemic component for prolonged remission.**
(PPTX)

**S1 Table. Preferred Reporting Items for Systematic Reviews and Meta-Analyses extension for Scoping Reviews (PRISMA-ScR) checklist.**
(DOCX)

**S2 Table. PICOS search strategy and sources for the review.**
(DOCX)

**S3 Table. Search strategy (MEDLINE).**
(DOCX)

**S4 Table. Characteristics of included studies.**
(DOCX)

**S5 Table. Combinations of numeric glycaemic parameters used to signify normoglycaemia for all definitions of remission.**
(DOCX)

**S6 Table. Possible interpretations of the 2009 report.**
(DOCX)

**S7 Table. Definitions of remission organised by unspecified remission, partial remission, complete remission, and prolonged remission.**
(DOCX)

## Author Contributions

**Conceptualization:** Mireille Captieux, Sarah Wild, Bruce Guthrie.

**Data curation:** Mireille Captieux, Regina Prigge, Sarah Wild, Bruce Guthrie.

**Formal analysis:** Mireille Captieux, Regina Prigge, Sarah Wild, Bruce Guthrie.

**Funding acquisition:** Mireille Captieux, Sarah Wild, Bruce Guthrie.

**Investigation:** Mireille Captieux, Regina Prigge, Sarah Wild, Bruce Guthrie.

**Methodology:** Mireille Captieux, Sarah Wild, Bruce Guthrie.

**Project administration:** Mireille Captieux.

**Supervision:** Sarah Wild, Bruce Guthrie.

**Visualization:** Mireille Captieux, Bruce Guthrie.

**Writing – original draft:** Mireille Captieux, Regina Prigge, Sarah Wild, Bruce Guthrie.

**Writing – review & editing:** Mireille Captieux, Regina Prigge, Sarah Wild, Bruce Guthrie.

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
