## [Editor Report · Decision Letter 0]

15 Jun 2020

Dear Dr Captieux, 

Thank you for submitting your manuscript entitled "Defining remission of type 2 diabetes in research studies: A systematic scoping review" for consideration by PLOS Medicine.

Your manuscript has now been evaluated by the PLOS Medicine editorial staff and I am writing to let you know that we would like to send your submission out for external peer review.

Kind regards,

Artur Arikainen,

Associate Editor

PLOS Medicine

---

## [Decision Letter · Decision Letter 1]

17 Jul 2020

Dear Dr. Captieux,

Thank you very much for submitting your manuscript "Defining remission of type 2 diabetes in research studies: A systematic scoping review" (PMEDICINE-D-20-02700R1) for consideration at PLOS Medicine. 

[LINK]

In light of these reviews, I am afraid that we will not be able to accept the manuscript for publication in the journal in its current form, but we would like to consider a revised version that addresses the reviewers' and editors' comments. Obviously we cannot make any decision about publication until we have seen the revised manuscript and your response, and we plan to seek re-review by one or more of the reviewers. 

We expect to receive your revised manuscript by Aug 07 2020 11:59PM. Please email us (plosmedicine@plos.org) if you have any questions or concerns.

We look forward to receiving your revised manuscript. 

Sincerely,

Artur Arikainen, 

Associate Editor 

PLOS Medicine

plosmedicine.org

1. Data Availability: At this stage, please make the raw extraction table available in a public repository or add the excel sheet to supplementary information file, as you suggest.

2. Abstract:

a. Please report your abstract according to PRISMA for abstracts, following the PLOS Medicine abstract structure (Background, Methods and Findings, Conclusions) - http://www.plosmedicine.org/article/info:doi/10.1371/journal.pmed.1001419.

b. Please avoid “consensus” for the 2009 report, here and throughout the text, as recommended by reviewer #1.

c. In the last sentence of the Abstract Methods and Findings section, please describe the main limitation(s) of the study's methodology.

4. Please use the "Vancouver" style for reference formatting, and see our website for other reference guidelines https://journals.plos.org/plosmedicine/s/submission-guidelines#loc-references. Citations should be in square brackets, and preceding punctuation.

5. Please avoid using italics for emphasis, eg. lines 66, 68, 274.

6. Please remove colons from the ends of section and subsection headings, eg “Search strategy and selection criteria”.

7. Please update your search, as it is now over 12 months old and relevant articles published in that time may have been missed.

8. Please remove the funding information on lines 167-170. This is taken from the online submission form instead.

9. Please present and organize the Discussion as follows: a short, clear summary of the article's findings; what the study adds to existing research and where and why the results may differ from previous research; strengths and limitations of the study; implications and next steps for research, clinical practice, and/or public policy; one-paragraph conclusion.

10. Please provide a URL for reference 162.

11. Please provide a copy of the completed PRISMA checklist. When completing the checklist, please use section and paragraph numbers, rather than page numbers.

--------

Comments from the reviewers:

Reviewer #1: The aims of this systematic review are very reasonable. Although the information distilled from the literature is unsurprising for those working in this area, the need for a widely accepted definition of remission of type 2 diabetes is not generally appreciated. Some inaccuracies require to be corrected. 

1. In the Abstract the statement that there is 'no consensus' requires to be rephrased. In the UK, the Primary Care Diabetes Society and the Association of British Clinical Diabetologists have published their consensus (Nagi et al 2019, Brit J Diabetes 19: 73-76; https://www.pcdsociety.org/news/details/acbd-and-pcds-release-joint-position-statement-on-remission-of-type-2-diabetes) and the 2009 Diabetes care consensus is still widely used. 

2. The ADA neither commissioned nor formally accepted the 2009 consensus document. Reference to this document requires to be adjusted throughout. it was a consensus statement from a group of experts, published in an ADA Journal. Even though the authors correctly state this (line 72-74), the description elsewhere is inaccurate. 

3. Line 42: the definition of normal fasting plasma glucose differs in the US (5.6mmol/l) and the UK (6.1 mol/l). The wording should be amended accordingly. 

4. Line 87. Undue emphasis appears to be given in the introduction to the 2009 expert group opinion. In a systematic review should one paper really monopolise the Introduction?

5. Line 129. Involvement of people with diabetes is important in respect of the critical matter of the ease of understanding and using a definition in the care of individuals. At the very least, the James Lind Alliance / DUK survey of the most important research questions in diabetes should be quoted (Finer et al; Diabetic Medicine 2018:35:862-870). This has major implications for the patient centred need to have HbA1c rather than fasting or OGTT plasma glucose criteria due to how Primary Care operates. Reference to the james Lind Alliance's identification of remission as the number 1 important research question would be appropriate. 

6. Line 305 "identifies' should be replaced with 'supports'. This point takes account of the almost completed consensus discussions between ADA, EASD and DUK which are almost complete. The present SR is timely in respect of likely publication of International consensus guidelines next year. 

7. Line 324: The word 'must' is inappropriate. For most people and most doctors, HbA1c provides a practical and appropriate basis for definition. However, haemoglobinopathies, iron deficiency states and pregnancy make it likely that an either or approach will be necessary. 

Reviewer #2: See attachment

Michael Dewey

Reviewer #3: See attachment

[LINK]

---

## [Decision Letter · Decision Letter 2]

26 Aug 2020

Dear Dr. Captieux,

Thank you very much for re-submitting your manuscript "Defining remission of type 2 diabetes in research studies: A systematic scoping review" (PMEDICINE-D-20-02700R2) for review by PLOS Medicine.

I have discussed the paper with my colleagues and the academic editor and it was also seen again by two reviewers. I am pleased to say that provided the remaining editorial and production issues are dealt with we are planning to accept the paper for publication in the journal.

[LINK]

We look forward to receiving the revised manuscript by Sep 02 2020 11:59PM. 

Sincerely,

Artur Arikainen, 

Associate Editor 

PLOS Medicine

plosmedicine.org

Requests from Editors:

1. Abstract:

a. Please summarise the numbers of each study type included (observational vs. interventional), and by regions or countries.

b. Please include a summary of inclusion criteria, eg. studies in the English language, with 100 or more participants.

c. Line 40: Please include percentage: “(47 definitions, 20%)”.

d. Please remove this sentence: “The direction of any bias is towards underestimating heterogeneity in defining remission.”

e. Please add one or two more limitations to the end of the Methods and Findings subsection.

f. Line 45: Begin with “We found that…”

g. Please include the principal source of funding.

h. Please move the registration information from lines 82-83 here.

2. Please remove spaces from within citation callouts, eg. “…and an ageing population [2,3].”, but keep the space between the text and the callout itself as is.

3. Introduction: Please mention briefly the position of the International Diabetes Federation on remission definition(s), if any.

4. Line 168: Please include day for the start date.

5. Fig 2 legend: Please replace the colour box with a description (eg. “Red indicates…”).

6. Please check the URL for reference 16.

7. Please remove the duplicate PRISMA checklist from the other Supporting Information file.

8. We would also recommend replacing your PRISMA checklist with the version specifically for scoping reviews: http://www.prisma-statement.org/Extensions/ScopingReviews

---------

Comments from Reviewers:

Reviewer #1: All points have been satisfactorily answered. The review is now accurately discussed and a useful contribution to the literature.

Reviewer #3: No further comments.

[LINK]

---

## [Editor Report · Decision Letter 3]

29 Sep 2020

Dear Dr Captieux, 

On behalf of my colleagues and the academic editor, Dr. Sanjay Basu, I am delighted to inform you that your manuscript entitled "Defining remission of type 2 diabetes in research studies: A systematic scoping review" (PMEDICINE-D-20-02700R3) has been accepted for publication in PLOS Medicine. 

PRODUCTION PROCESS

Before publication you will see the copyedited word document (within 5 busines days) and a PDF proof shortly after that. The copyeditor will be in touch shortly before sending you the copyedited Word document. We will make some revisions at copyediting stage to conform to our general style, and for clarification. When you receive this version you should check and revise it very carefully, including figures, tables, references, and supporting information, because corrections at the next stage (proofs) will be strictly limited to (1) errors in author names or affiliations, (2) errors of scientific fact that would cause misunderstandings to readers, and (3) printer's (introduced) errors. Please return the copyedited file within 2 business days in order to ensure timely delivery of the PDF proof. 

If you are likely to be away when either this document or the proof is sent, please ensure we have contact information of a second person, as we will need you to respond quickly at each point. Given the disruptions resulting from the ongoing COVID-19 pandemic, there may be delays in the production process. We apologise in advance for any inconvenience caused and will do our best to minimize impact as far as possible.

PRESS

PROFILE INFORMATION

Thank you again for submitting the manuscript to PLOS Medicine. We look forward to publishing it. 

Best wishes, 

Artur Arikainen, 

Associate Editor 

PLOS Medicine

plosmedicine.org